# Activity of *Colocasia esculenta* (Taro) Corms against Gastric Adenocarcinoma Cells: Chemical Study and Molecular Characterization

**DOI:** 10.3390/ijms25010252

**Published:** 2023-12-23

**Authors:** Tiziana Esposito, Simona Pisanti, Luciano Mauro, Teresa Mencherini, Rosanna Martinelli, Rita Patrizia Aquino

**Affiliations:** 1Department of Pharmacy, University of Salerno, 84084 Fisciano, SA, Italy; tesposito@unisa.it (T.E.); lmauro@unisa.it (L.M.); aquinorp@unisa.it (R.P.A.); 2UNESCO Chair Salerno Plantae Medicinales Mediterraneae, University of Salerno, 84084 Fisciano, SA, Italy; 3Department of Medicine, Surgery and Dentistry ‘Scuola Medica Salernitana’, University of Salerno, 84081 Baronissi, SA, Italy; spisanti@unisa.it; 4Giardino della Minerva, 84121 Salerno, SA, Italy

**Keywords:** *Colocasia esculenta*, gastric adenocarcinoma cells, apoptosis, cell cycle, molecular mechanisms, pure molecules

## Abstract

*Colocasia esculenta* (L.) Schott is a tuberous plant, also known as taro, employed as food worldwide for its renowned nutritional properties but also traditionally used in several countries for medical purposes. In this study, methanolic extracts were prepared from the corms and leaves of *Colocasia*, subsequently fractionated via molecular exclusion chromatography (RP-HPLC) and their anti-tumor activity assessed in an in vitro model of gastric adenocarcinoma (AGS cells). Vorm extract and isolated fractions II and III affected AGS cell vitality in a dose-dependent manner through the modulation of key proteins involved in cell proliferation, apoptosis, and cell cycle processes, such as caspase 3, cyclin A, cdk2, IkBα, and ERK. To identify bioactive molecules responsible for anti-tumoral activity fractions II and III were further purified via RP-HPLC and characterized via nuclear magnetic resonance (NMR) and electrospray mass spectrometry (ESI-MS) techniques. The procedure enabled the identification of ten compounds including lignans and neolignans, some isolated for the first time in taro, uncommon megastigmane derivatives, and a gallic acid derivative. However, none of the isolated constituents showed efficacy equivalent to that of the fractions and total extract. This suggests that the whole *Colocasia* phytocomplex has intriguing anti-tumor activity against gastric cancer.

## 1. Introduction

*Colocasia esculenta* (L.) Schott, belonging to the family Araceae, is commonly known as taro. This herbaceous perennial plant features a robust, short caudex and erect stems that can reach a height of 1–2 m. The leaves are typically large and elongated, often referred to as “elephant head” leaves. Additionally, the plant possesses a central starchy corm, which can weigh up to 4 kg and has a cylindrical or spherical shape with a diameter of approximately 30 cm [1]. Taro holds a significant historical position as one of the oldest cultivated crops globally. Its origin can be traced back to the Southeast Asian, Australian, and Papua New Guinean regions. Today, it is widely cultivated for its starchy edible corms in tropical to temperate regions worldwide [2,3] and is processed into various food products such as flour, pasta, canned goods, and bars. Nutritional properties are related to high starch content, comprising 70–80% of their dry weight, as well as to the small size of the starch granules (1–4 mm in diameter) that are easily digestible. This characteristic makes taro an ideal choice for individuals with dietary restrictions, including those with allergies and gluten intolerance, especially among children. Moreover, taro corms are a rich source of health-promoting compounds, calcium, iron, vitamins, and minerals [1,4].

Besides the corms, the leaves also offer nutritional benefits [5]. They contain proteins, minerals, secondary metabolites, and fibers, and are commonly used in culinary preparations such as sauces, soups, and stews [1]. Interestingly, even after cooking, these vegetable preparations retain the bioactive molecules, suggesting their potential inclusion in daily dietary habits. Despite its natural adaptation to tropical conditions, taro is cultivated in the Mediterranean regions and Europe [6]. Today, in Southern Italy, *C. esculenta* is predominantly employed as an ornamental plant to embellish fountains and ponds.

As the medicinal uses, knowledge of *Colocasia* dates to the distant Middle Ages, and the plant is mentioned in treatises on medicinal herbs associated with the renowned mediaeval Salerno Medical School [2]. Traditionally, taro is used in Asia, West Africa, Central and South America, Turkey, and Japan to treat many conditions such as asthma, arthritis, diarrhea, hemorrhage, diabetes, hypertension, neurological and skin problems [4]. Taro is also utilized as a tonic, carminative, diuretic, and expectorant, to treat body aches and to stimulate the immune system among others. Recent studies have confirmed the abundance in taro of bioactive compounds that exhibit numerous beneficial properties in various pathophysiological conditions, such as anti-oxidant, anti-inflammatory, immunomodulatory, hepatoprotective, anti-diabetic, anti-hypercholesterolemic, anti-microbial, anti-viral, and anti-cancer [7,8,9,10,11], according to some of its acknowledged traditional uses. Anti-cancer and anti-metastatic properties have been demonstrated for water-soluble extracts of corms, identifying proteins and polysaccharides such as tarin and taro-4-I as responsible for the activity [12,13,14].

Stomach cancer is the fifth most common cancer, with over one million cases annually, and the fourth leading cancer killer worldwide. Its prognosis is adverse, as diagnosis in about 80% of cases occurs in advanced stages, when the cancer has already spread to lymph nodes or other organs. A chronic inflammatory state caused by *Helicobacter pylori* infection, a diet rich in salty, smoked, or preserved foods with salt and nitrites, and pathological conditions such as obesity are associated with an increased production of TNFα, IL-6, and other proinflammatory cytokines that can promote gastric carcinogenesis. The best preventive measures include behavioral changes in diet, increasing fresh fruit and vegetable consumption, reducing the intake of salt and preserved foods, and lifestyle changes [15].

Given its high nutritional value and beneficial pharmacological properties, the present study aimed to deeply investigate in vitro anti-tumor activity of alcoholic extracts from corms and leaves of *Colocasia* on human gastric adenocarcinoma (AGS) cells. The effects of extracts on AGS cell vitality were assessed and the active corm extract (CCM) was subjected to bioassay-oriented fractionation. The main secondary metabolites from bioactive fractions II and III were isolated and identified. The involvement of whole extract and fractions II and III in inducing apoptosis and modulating key proteins of cell proliferation, apoptosis, and cell cycle processes was investigated.

## 2. Results 

### 2.1. Production of the Methanolic Extract of Colocasia Corms and Leaves and Cytotoxicity in AGS Cells

Dried corms and leaves of *Colocasia* were separately subjected to solid–liquid extraction using solvents of increasing polarity (*n*-hexane, chloroform, and methanol), obtaining two dry methanol extracts (CCM and CLM). At first, the cytotoxic activity of such methanolic extracts was evaluated by means of the MTT cell viability assay on AGS cells. The cells were treated separately with the CLM and CCM extract of *Colocasia* in a concentration range between 6.25 and 400 μg/mL. Only the CCM extract showed dose-dependent cell viability inhibitory activity against AGS cells at all tested concentrations, while the CLM extract did not display any significant effect. The IC_50_ of CCM extract, calculated on the basis of the maximum observed effect, was 21.86 µg/mL (Figure 1A).

### 2.2. Studies on Cytotoxicity of Colocasia Corm Fractions in AGS Cells

In order to potentially find if there are single molecules responsible for the cytotoxic effect of CCM or if it depends on the phytoextract as a whole, the bioactive CCM extract was subjected to fractionation using molecular exclusion chromatography, resulting in four major fractions (I–IV). The investigation of the inhibitory effect of AGS cell viability was then performed on the obtained CCM fractions (I, II, III, and IV) via an MTT assay. The cells were treated separately with the four fractions at concentrations ranging from 6.25 to 400 μg/mL. The fractions most active in inhibiting the viability of AGS cells were II and III, which showed an IC_50_ of 44.74 μg/mL and 51.76 μg/mL, respectively (Figure 1B).

### 2.3. Colocasia Methanolic Extract and Fractions Induce Apoptosis in AGS Cells Modulating the Expression of Key Proteins Involved in Cell Proliferation, Cell Cycle, and Apoptosis Processes

The cell cycle distribution of AGS was assessed after treatment with two selected cytotoxic concentrations (100 and 200 μg/mL) of CCM and its active fractions II and III in order to verify the exact nature of the antiproliferative effect of *Colocasia*. We did not observe a significant cytostatic effect exerted by either CCM extract or fractions II and III, as assessed via flow cytometry (Figure 2A) even if immunoblot analysis revealed a small, albeit significant induction of cyclin A and cdk2 (only at the highest dose tested of 200 μg/mL), which are fundamental for the passage into the S phase (Figure 2B). Noteworthily, the cell cycle profile showed a dose-dependent increase in the percentage of sub-G1 cells with CCM as well as with fractions II and III, suggestive of apoptosis induction rather than of an antiproliferative cytostatic effect.

To confirm this hypothesis and assess whether or noy the inhibition of cell viability was caused by an increased mortality of AGS cells treated with the extract and its active fractions (II and III), a more sensitive apoptosis assay was subsequently performed via flow cytometry multiparametric analysis with the double labeling of cells with anti-human Annexin-V antibody and propidium iodide. AGS cells were treated separately with CCM and fractions II and III at selected active concentrations of 50 and 100 μg/mL. The Annexin/PI test showed a significant increment in the percentage of dead cells (Annexin+ and Annexin+/PI+) for both CCM and its active fractions II and III at all concentrations tested (Figure 3A). The proapoptotic effect was confirmed by evaluating the expression of key proteins involved in apoptosis by means of Western blot analysis. AGS cells were treated separately with CCM whole extract and fractions II and III at concentrations of 100 and 200 μg/mL for 24 h. Both fractions (II and III) mediate the activation of Caspase 3, with a decrease in the expression of the whole form and an increase in the expression of the active cleaved form. An increase in the phosphorylation of IkBα (nuclear factor of kappa light polypeptide gene enhancer in B-cells inhibitor, alpha) was also observed, the degradation of which induces the activation of NF-κB (nuclear factor kappa-light-chain-enhancer of activated B cells), a transcription factor essential for the regulation of pro-inflammatory protein expression, which can also play a pro-apoptotic role. Furthermore, the results show a decrease in the phosphorylation of ERK (extracellular signal-regulated-kinase), which plays an important role in cell proliferation and tumor progression (Figure 3B).

### 2.4. Chemical Profile and Biological Activity of Colocasia Constituents

Based on the previous results on AGS cells, the next step was to identify the constituents of cytotoxic fractions II and III from the CCM extract. The main secondary metabolites were isolated via RP-HPLC, and their chemical structure was determined via NMR and MS data in comparison with those reported in the literature. The isolated CCM compounds were identified as lignans ((lariciresinol-4-*O*-β-d-glucopyranoside (**1**); lariciresinol-9-*O*-β-d-glucopyranoside (**2**); (+)-isolariciresinol (**6**), and (+)-lariciresinol (**9**)), a neolignan (americanol A (**10**)), glucosidic neolignans (dehydrodiconiferyl alcohol-4-*O*-β-d-glucopyranoside (**5**) and dehydrodiconiferyl alcohol-9-*O*-β-d-glucopyranoside (**7**)), megastigmane derivatives (byzantionoside B 9-*O*-*α-*l-rhamnopyranosyl-(1″→2′)-β-d-glucopyranoside (**3**) and byzantionoside B (**4**)), and a phenolic acid (3,4,5-tri-O-methyl gallic acid (**8**)) (Figure 4). The content of compounds **6**, **7**, and **10** in CCM extract was determined via high-performance liquid chromatography with diode-array detection (HPLC-DAD) using a direct calibration method and isolated compounds for building calibration curves. The results show that (+)-isolariciresinol (**6**), dehydrodiconiferyl alcohol-9-*O*-β-D glucopyranoside (**7**), and americanol A (**10**) accounted for 1.50, 1.70, and 1.65 % (*w*/*w*), respectively, of CCM and were selected as chemical markers of the whole extract.

All the isolated molecules were tested for biological activity in AGS cell vitality using an MTT assay. Most of the molecules did not show any significant inhibitory activity in the range of concentrations tested (5–500 μg/mL). The chemical markers displayed an inhibitory effect only at the highest concentrations (% of inhibition vs. control at the 250 μg/mL dose: **6**, 22%; **7**, 22%; **10**, 35%). Notably, these concentrations were several orders higher than the quantity in μg of each molecule found in 100 μg of the total CCM extract reported above.

## 3. Discussion

Corms and leaves of *Colocasia* are cooked and eaten by the populations of tropical and subtropical areas as starchy food and vegetables, as well as being used in folk medicine. Interestingly, references to *Colocasia* in texts by Italian mediaeval physicians belonging to the Medical School of Salerno [16] indicate ancient knowledge of the plant even in southern Europe and suggest its potential medicinal use in those regions [2]. For this study, the corms and leaves were collected from the plants which still grow in the “Giardino della Minerva”, a terraced garden established in the 14th century in Salerno, a city in southern Italy, which can be considered one of the oldest botanical gardens in Europe. Recent studies [14] confirmed the presence of bioactive molecules in the different parts of the plant suggesting interesting health applications and pharmacological properties, such as anti-oxidant, anti-inflammatory, anti-diabetic, anti-hyperlipidemic, immunomodulatory, anti-tumor and anti-metastatic. Such molecules, acting in synergy, can help to prevent tumor growth on multiple fronts by modulating the inflammatory state as well as exerting a direct cytotoxic action on the tumor mass.

Considering the main use of *Colocasia* as food, with a potential direct impact on the gastrointestinal system, the anti-tumor efficacy of the methanolic extract obtained from dried leaves and corms of *Colocasia* was evaluated in an in vitro model of gastric adenocarcinoma. To the best of our knowledge, the present study is the first to investigate the anti-tumor potential of an organic extract of *Colocasia* in gastric adenocarcinoma.

Taro crude and ethanolic extracts have been reported to have preclinical promising anti-tumor activity in T cell leukemia, colon, breast, and prostate cancer models, inhibiting cell proliferation and drastically reducing metastases as observed in two syngeneic murine models of triple-negative breast cancer metastases to the lung and heart [12,17,18,19]. Anti-metastatic efficacy is derived both from the direct action of the inhibition of tumor cell migration via the downregulation of the COX1-2/PGE pathway and through indirect mechanisms with the activation of the anti-tumor immune response.

Most studies investigating the anti-tumor properties of *Colocasia* focused on tarin and taro-4-I polysaccharide as responsible for the anti-tumor effects observed. Tarin, a lectin protein, was reported to inhibit the proliferation of breast, hepatocellular carcinoma, and glioblastoma [20,21]; Taro-4-I polysaccharide inhibited mouse lung metastases in murine melanoma cells [13].

In the present study, we observed that *Colocasia* corms and not methanolic extract from leaves showed a significant cytotoxic effect on human gastric adenocarcinoma cells mediated by apoptosis through the increased activity of caspase-3. A previous report observed that a soluble extract of *poi*, a starchy paste of cooked and fermented taro typical of Hawaii, elicited morphological changes suggestive of apoptosis in rat colon cancer cells [17]. However, our work is the first to demonstrate, through specific assays both at the cellular and molecular level, the involvement of apoptosis induction in the anti-tumor properties of *Colocasia* extract and its active fractions, using human cells, which strengthens the significance of the effects observed. It is quite interesting that the antiproliferative effects noted are not accompanied by a substantial change in the distribution of cells among the cell cycle phases, excluding a mere cytostatic effect, even if a slight but significant increment in the expression of cyclin A and cdk2 was reported. It is known that cyclin A and cdk2 have a complex role in cancer and are frequently mutated and overexpressed in several tumor types, including stomach adenocarcinoma [22]. Interestingly, the hypothesis of a tumor suppressor function for the cyclin A-cdk2 complex has been put forward in that, by phosphorylating the p53 protein, it stimulates its binding to DNA and the subsequent activation of downstream mechanisms, including apoptosis [23].

It is noteworthy that several reports have already demonstrated that the beneficial anti-tumor action of *Colocasia* corms is not due to unspecific cytotoxicity. Indeed, both studies on crude extract as well as on tarin and taro-4-I have reported no cytotoxic effects on spleen cells in vitro and in vivo [14]. The observed immunomodulatory effect, together with the anti-inflammatory efficacy, could even overall contribute to the anti-cancer properties of corms.

Both the total extract of *Colocasia* corms and its active fractions inhibited phosphorylation and thus the activation of ERK, which plays an important role in cell proliferation and tumor progression. Indeed, it is known that the deregulation of MAPK downstream signaling pathways plays a crucial role in the control of cell proliferation, migration, survival, metabolism, and differentiation. Therefore, blocking this pathway results in an effective anti-tumor effect [24]. Our results suggest that *Colocasia* modulates the ERK signaling pathway, inhibiting cell proliferation, and inducing apoptosis. Increased phosphorylation of IkBα was also observed in our model. Degradation of IkBα induces the activation of NF-κB, a transcription factor that regulates a variety of cellular processes, including inflammation, immune response, cell proliferation, and death. NF-κB is a dimeric protein normally found in the cytoplasm, where it is maintained in an inactive state by binding to IkB. The phosphorylation of IkB by the kinase IKK results in its degradation at the proteasome and in the release of NF-κB, which migrates into the nucleus where it activates the expression of target genes. While NF-κB is most commonly involved in the suppression of apoptosis through the transactivation of anti-apoptotic gene expression, it can also promote programmed cell death in response to specific death signals. In relation to the role of NF-κB in cancer, it has been reported to be pro-apoptotic in the anti-tumor action of cisplatin in head and neck squamous cell carcinoma, in glioblastoma in TRAIL-induced cell death, and in adenocarcinoma cells infected with retroviruses or stimulated with TNF-α [25,26,27,28]. Overall, NF-κB may act synergistically with ROS in inducing apoptosis in cancer cells by naturally occurring molecules, as recently observed in AGS cancer cells treated with lutein [29]. Therefore, *Colocasia* corm extract could act through the NF-κB pathway to promote apoptosis.

It is well established that diet is a risk factor for the development of cancer of the gastrointestinal tract. Several health-protective benefits have been associated with a diet rich in phytochemicals that have been shown to modulate critical cellular pathways. Few previous works have reported the chemical components of taro corms [30,31,32,33]. Fatty acid derivatives and phenolic compounds (catechins, flavonoids, and phenolic acids) were identified in polar extracts. Kim et al. [34] isolated lignans, such as americanol A (**10**), from a methanolic extract of tuber barks. We proceeded to an investigation of the methanolic extract and its fractions assessing the composition and properties, providing valuable information about its secondary metabolites and potential bioactive components. Compounds **1**, **2**, **6**, and **9** are lignans, and compounds **10**, **5**, and **7** are neolignans, both dimeric polyphenols exhibiting a wide range of structural variations and known for their diverse biological activities, primarily attributed to their ability to reduce oxidative stress and inflammatory processes [35]. The main structural difference between the two groups of plant metabolites is the type of bond between the phenylpropanoid monomers from which they are formed. The lignan skeleton contains a linkage between the 8 and 8′ carbons, while in neolignans, different carbons are involved or an oxygen-ether bond is present [36]. Different parts of higher plants, mainly roots, rhizomes, hardwood, seeds, and fruits may contain lignans even if in small quantities [35,37]. Despite the wide distribution in higher plants, lignan (**1**, **2**, **6**, and **9**) and neolignan (**5** and **7**) derivatives were identified in taro corms for the first time in the present research.

Furthermore, the isolated megastigmane **3** is a terpenoid that is rather rare in nature due to the presence of a glycosidic bond, 1″ → 2′, between rhamnose and glucose residues of the sugar moiety. Typically, glycosidic megastigmanes commonly have a 1″ → 6′ linkage in the disaccharide unit. The identification of **3** has been reported in a limited number of plant species [38,39,40,41] including *Epipremnum pinnatum* [42], which, like *Colocasia*, belongs to the Araceae family.

To ensure the production of a standardized extract and verify the reproducibility of the extraction method and its associated biological effects, the main compounds in the CCM extract were identified as (+)-isolariciresinol (**6**), dehydrodiconiferyl alcohol-9-*O*-β-d-glucopyranoside (**7**), and americanol A (**10**), and quantified via HPLC-DAD analysis. The cytotoxic activity of the same or structurally related lignans and neolignans on a wide range of cell lines is reported in the literature [43,44,45,46]. (+)-isolariciresinol and dihydrodehydrodiconiferyl alcohol-9-*O*-β-d-glucopyranoside resulted to be cytotoxic against oral adenosquamous, hepatoma, ovarian and colorectal carcinoma, and breast and lung adenocarcinoma cells with IC_50_ values ranging from 57.8 to 95.18 μg/mL.

Nevertheless, the demonstrated anti-tumor potential of *Colocasia* against gastric adenocarcinoma cannot be attributed to one of these isolated metabolites. As often happens, many molecules, even minor ones, may contribute to the effectiveness of the whole plant extract with synergistic effects and positive interactions [47,48,49].

In conclusion, in this study, we demonstrated the potential anti-tumor activity of *Colocasia* corm extract and its fractions on gastric adenocarcinoma cells. Mechanistically, taro extract and some of its fractions induced the activation of apoptosis markers and affected the phosphorylation status of proteins that play crucial roles in cell proliferation and tumor progression. Further chemical analysis of the active fractions led to the identification, for the first time, of bioactive phenolics and uncommon terpenoids providing valuable information on taro composition. Individual testing of the major compounds on AGS cells showed weak cytotoxic effects, suggesting the value of the whole corm where different bioactive molecules work in synergy. The obtained results enhance the understanding of a dietary strategy based on edible taro in potentially protecting and lowering the risk of gastric cancer and provide convincing evidence of its in vitro anti-tumor activity on AGS cells, which may be a key step in the development of a new nutraceutic/chemopreventive agent.

## 4. Materials and Methods

### 4.1. Chemicals and Reagents

Analytical-grade solvents, including n-hexane, chloroform, and methanol, were used for the extraction and isolation procedures. The MTT reagent (3-(4,5-dimethylthiazol-2-yl)-2,5-diphenyltetrazolium bromide), HPLC-grade methanol, and deuterated methanol (CD_3_OD) were purchased from Sigma-Aldrich (Merck KGaA, Darmstadt, Germany). Cerium sulphate, used in the study, was also obtained from Sigma-Aldrich. HPLC-grade water with a resistivity of 18 MΩ was prepared using a Milli-Q_50_ purification system from Millipore Corp. (Bedford, MA, USA). The Sephadex LH-20 resin employed in the chromatography procedure was purchased from Supelco (Merck KGaA, Darmstadt, Germany). Furthermore, dimethyl sulfoxide (DMSO), used for solubilizing the extract and molecules during bioassays, was obtained from VWR Chemicals (VWR International Srl, Milano, Italy).

### 4.2. General Experimental Procedures

A Bruker DRX-600 NMR spectrometer, operating at 599.19 MHz for ^1^H and 150.858 MHz for ^13^C, using the TopSpin 3.2 software package, was used for the NMR experiments in CD_3_OD. The chemical shifts are expressed in *δ* (parts per million), referring to the solvent peaks δ_H_ 3.31 and δ_C_ 49.05 for CD_3_OD, with coupling constants, *J*, in Hertz. Conventional pulse sequences have been used for ^1^H-^1^H DQF-COSY, ^1^H-^13^C HSQC, and HMBC experiments [50]. ESI-MS experiments [51] were conducted using a Finnigan LC-Q Deca spectrometer (Thermoquest, San Jose, CA, USA), which was equipped with Xcalibur 3.1 software (Thermoquest, San Jose, CA, USA). Molecular exclusion chromatography was performed on Sephadex LH-20. HPLC separations were performed with a Waters 515 series pump system, equipped with a Waters R401 refractive index detector and a Rheodyne injector (100 μL loop), using Synergy Fusion C_18_ (250 × 10 mm i.d., 4 μm, Phenomenex, Torrance, CA, USA), at a flow rate of 2 mL/min. Thin-layer chromatography (TLC) was performed with silica gel plates 60F-254 (Delchimica, Naples, Italy) and cerium sulphate spray reagent, and a UV lamp (254 and 366 nm) was used to visualize the spots. Quantitative analysis of the extract was performed with Agilent 1100 HPLC with G1312A Binary Pump, G1315B Diode Array Detector (DAD), and G1328B Injector (Agilent Technologies, Santa Clara, CA, ISA).

### 4.3. Plant Material

Leaves and corms of *C. esculenta* were collected in July 2021, at Giardino della Minerva (40°40′52″ N 14°45′13.39″ E), Salerno (Italy). The plant material was identified by one of the authors (T. Mencherini). A voucher specimen (CECL_21) was deposited at the Department of Pharmacy, University of Salerno. The leaves (1.3 kg) were air-dried, for five days, at a controlled temperature (25 ± 2 °C, RH 40%), while the corms (2.5 kg) were freeze-dried (Lyovapor L-200, BUCHI Italia s.r.l., Cornaredo, Italy) for five days until the weight of the samples was constant. The drying yield was determined gravimetrically (Denver Instruments-PK-601, max 60000 gd = 0.1 g; +15/30 °C, Gemini b.v., Apeldoorn, The Netherlands) and expressed as a weight percentage of dry matter with respect to the total amount of fresh initial biomass and results of 12.8% *w*/*w* for the leaves and 15.9% *w*/*w* for the corms. The dried corms and leaves were separately pulverized with a GM 200 knife mill (Retsch GmbH, Haan, Germany), in cut mode (2000 rpm × 10″ for corms and 1000 rpm × 2″ for leaves), and sieved with a Retsh vibrating sieve (Retsch GmbH, Haan, Germany) for 10′ with Amplitude 1.45 to obtain powders with a homogeneous particle size distribution (450 µm).

### 4.4. Extraction Procedures

An aliquot of grounded *Colocasia* corms (CC, 375.1 g) and leaves (CL, 151.9 g) was separately defatted with *n*-hexane and chloroform and extracted at room temperature with methanol (M) (3 × 2 L). After the solvent evaporation at 40 °C under vacuum (Heidoplh Hei-VAP Value Digital, Schwabach, Germany) the dried residues CCM and CLM were obtained. The extraction yield, determined gravimetrically (Equilibrium Denver Instruments-PK-201, max 2400 gd = 0.1 g; +15/30 °C), was expressed as the weight percentage of the dry matter compared to the total amount of the plant powder, and it was 5.3 and 21.0% *w*/*w* for CCM and CLM, respectively.

### 4.5. Isolation Procedure of Compounds ***1***–***10***

A portion of CCM (2.8 g) was fractionated on a Sephadex LH-20 column (1 m × 5 cm) using MeOH as eluent, at a flow rate of 1 mL/min. The fractions, 7 mL each, were collected and combined into four main groups (I–IV) after TLC analysis (Si-gel, *n*-BuOH AcOH-H_2_O (60:15:25)). The effect of fractions I (580.36 mg), II (450.24 mg), III (543.18 mg), and IV (474.34 mg) on AGS cell viability was investigated.

Bioactive fractions II and III were purified via RP-HPLC with a refractive index detector on a Synergi Fusion C_18_ column (250 × 10 mm i.d., 4 μm, Phenomenex, Torrance, CA, USA), at a flow rate of 2 mL/min.

Fraction II (318.8 mg) was purified with the elution solvent MeOH/H_2_O 3.5:6.5 *v*/*v*, yielding the following compounds **1** (0.6 mg, t_R_ = 35 min) and **2** (0.6 mg, t_R_ = 50 min). Fraction II (20.0 mg) was also purified using a mixture of MeOH/H_2_O 5:5 *v*/*v* as a mobile phase, isolating the compounds **3** (0.7 mg, t_R_ = 19 min) and **4** (0.9 mg, t_R_ = 23 min).

Fraction III (358.2 mg) was purified with the elution solvent MeOH/H_2_O, 3.5:6.5 *v*/*v* to obtain the following compounds: **5** (1.0 mg, t_R_ = 35 min), **6** (0.9 mg, t_R_ = 43 min), and **7** (1.2 mg, t_R_ = 62 min). Fraction III (58.3 mg) was also purified using a mixture of MeOH/H_2_O 5:5 *v*/*v* as a mobile phase, isolating compounds **8** (0.2 mg, t_R_ = 25 min), **9** (1.2 mg, t_R_ = 20 min), and **10** (2.0 mg, t_R_ = 23 min).

### 4.6. Spectroscopic Data of Compound ***1***–***10***

lariciresinol-4-*O*-*β*-d-glucopyranoside (**1**). NMR and optical rotation data were consistent with those previously reported [52]. ESI-MS (positive mode), *m*/*z* 523.0 [M + H]^+^.

(+)-lariciresinol-9-*O*-*β*-d-glucopyranoside (**2**). NMR and optical rotation data were consistent with those previously reported [53]. ESI-MS (negative mode), *m*/*z* 521.0 [M − H]^−^.

byzantionoside B 9-*O*-*α*-l-rhamnopyranosyl-(1″→2′)-*β*-d-glucopyranoside (**3**). NMR data were consistent with those previously reported [54]. ESI-MS (positive mode), *m*/*z* 541.0 [M + Na]^+^.

byzantionoside B (**4**). NMR and optical rotation data were consistent with those previously reported [55]. ESI-MS (positive mode), *m*/*z* 373.5 [M + H]^+^.

dehydrodiconiferyl alcohol-4-*O*-*β*-d-glucopyranoside (**5**). NMR and optical rotation data were consistent with those previously reported [52]. ESI-MS (positive mode), *m*/*z* 521.1 [M + H]^+^.

(+)-isolariciresinol (**6**). NMR data were consistent with those previously reported [56]. ESI-MS (positive mode), *m*/*z* 361.0 [M + H]^+^.

dehydrodiconiferyl alcohol-9-*O*-*β*-d-glucopyranoside (**7**). NMR data were consistent with those previously reported [57]. ESI-MS (negative mode), *m*/*z* 521.0 [M − H]^*^

*3*,4,5-tri-*O*-methyl gallic acid (**8**). NMR data were consistent with those previously reported [58]. ESI-MS (negative mode), *m*/*z* 213.0 [M+H]^+^.

(+)-lariciresinol (**9**). NMR data agreed with those previously reported [53]. ESI-MS (positive mode), *m*/*z* 360.0 [M + H]^+^.

americanol A (**10**). NMR and optical rotation data were consistent with those previously reported [59]. ESI-MS (positive mode), *m*/*z* 331.1 [M + H]^+^.

NMR spectra are included as Appendix A.

### 4.7. Quantitative HPLC Analysis of CCM

Quantitative analysis was performed via HPLC-DAD [60] using a Synergi Fusion C_18_ column (80 Å, 250 mm × 4.6 mm i.d., 4 μm, Phenomenex, Torrance, CA, USA), at a flow rate of 1 mL/min, with an injection volume of 20 µL. Elution was performed in a gradient. The mobile phase was H_2_O + 1% HCOOH (solvent A) and MeOH + 1% HCOOH (solvent B). The selected elution gradient was as follows: 0 → 3 min, 10% B; 3 → 10 min, 35% B; 10 → 20 min, 35% B; 20 → 30 min, 50% B; 30 → 47 min, 60% B; 47 → 53 min, 100% B. Analyses were performed in triplicate. The absorbance was monitored with a DAD detector, model G-1315A, set at λ values of 280 and 270 nm. Four different solutions containing the compounds (+)-isolariciresinol (**6**), dihydrodehydroconiferyl alcohol-9-*O*-β-d-glucopyranoside (**7**), and americanol A (**10**), solubilized in MeOH/H_2_O 35% (*v/v*), were prepared in a concentration range between 0.25 and 0.125 mg/mL. CCM was solubilized and analyzed under the same conditions as were the isolated compounds. The peak associated with each compound within the extract was identified via a retention time comparison and confirmed through the co-injection of CCM with each isolated compound (**6**, **7**, and **10**). The calibration curves of the compounds dihydrodehydroconiferyl alcohol-9-*O*-β-d-glucopyranoside (**7**), americanol A (**10**), and (+)-isolariciresinol (**6**) (determined at λ 280 nm) show a linear correlation between the concentration and peak area of the injected compounds. The regression equations are as follows: y = 610.69x + 6.85 and R^2^ = 0.9984 for **6**; y = 599.12x − 5 and R^2^ = 0.9999 for **7**; y = 779.03x − 8.9 and R^2^ = 0.9983 for **10**, where y is the peak area and x the concentration of the tested compound.

### 4.8. Cells

Gastric adenocarcinoma cells (AGS) (ATCC CRL-1739) were obtained from ATCC (Manassas, VU, USA). Cells were grown in RPMI medium (GIBCO, Grand Island, NY, USA) with the following supplements: 10% heat-inactivated fetal bovine serum (Euroclone, Pero, Italy), 2 mM of L-glutamine, 50 ng/mL of streptomycin, and 50 units/mL of penicillin, all purchased from Sigma-Aldrich (St. Louis, MO, USA). Cells were maintained at 37 °C in a humidified 5% CO_2_ atmosphere.

### 4.9. Determination of Cell Viability via MTT Assay

AGS cells (6 × 10^3^/well) were cultured at 37 °C for 24 h into 96-well plates before adding extracts, fractions, or molecules at the indicated concentrations. Extracts, fractions, and molecules were prepared in DMSO, and the highest concentration assessed was selected to keep the DMSO concentration on cells below 0.5%, to avoid unwanted effects on cell viability. The treatment was stopped after 24 h, adding 10 µL of MTT solution (5 mg/mL, in water) (Thermo Fisher Scientific, MA, USA) for the last 2 h. The formazan salts were dissolved overnight in 100 µL of solubilization solution (10% Triton X-100 and 0.1 N HCl in isopropanol). Absorbance intensity was measured on a TECAN Infinite M200Pro microplate reader (Tecan Trading AG, Männedorf, Switzerland) at 570 nm with a reference wavelength of 650 nm. Values from treated cells were normalized against those of the control DMSO tested in the concentration range from 0.4 to 6.25 × 10^−3^%. All experiments were repeated at least three times in technical replicates, and cell vitality was expressed as a percentage versus that of the untreated control cells (100%).

### 4.10. Cell Cycle Analysis

AGS cells were cultured in 100 mm dishes at a density of 2 × 10^4^ cells/cm^2^ for 24 h and then treated with selected concentrations of extracts and fractions. Control cells were treated with 0.2% DMSO, which is the maximum concentration of DMSO achieved in the treatments. After 24 h of treatment, the cells were collected, fixed in 70% ethanol, and kept at −20 °C at least overnight. Following PBS washing, propidium iodide (PI; 50 μg/mL) in PBS containing 100 U/mL of DNase-free RNase was added to the cells for 15 min at room temperature. The cells were then acquired using a FACSVerse flow cytometer (BD Biosciences, San Jose, CA, USA); 20,000 events, corrected for debris and aggregate populations, were collected. Analysis was performed with the FlowJo v.10.6.2 (BD Biosciences) cell cycle tool.

### 4.11. Apoptosis Analysis

Apoptosis was assessed using Annexin V, FITC Apoptosis Detection Kit (Dojindo EU GmbH, Munich, Germany). Cells were plated and treated as described above. Control cells were treated with 0.1% DMSO, the maximum DMSO concentration in the treatments. Then, they were resuspended in Annexin V binding buffer and stained with Annexin V-FITC for 20 min followed by PI staining at room temperature for an additional 15 min in the dark. The cells were acquired using an FACSVerse flow cytometer (BD Biosciences) within 1 h after staining. At least 20,000 events were collected, and the data were analyzed using BD FACSuite analysis software v1.0.

### 4.12. Western Blot

Cells were grown and treated as described above. Control cells were treated with 0.2% DMSO, which is the maximum concentration of DMSO reached in the treatments. At the end of treatment, cells were washed with PBS, harvested, and lysed in ice-cold RIPA lysis buffer (50 mM Tris-HCl, 150 mM NaCl, 0.5% Triton X-100, 0.5% deoxycholic acid, 10 mg mL^−1^ leupeptin, 2 mM phenylmethylsulfonyl fluoride and 10 mg mL^−1^ aprotinin). Following cell debris removal via centrifugation (13,000× *g* for 20 min at 4 °C), total protein quantity was estimated. About 30 µg of proteins were loaded on 10–12% SDS–polyacrylamide gels under reducing conditions and then transferred to nitrocellulose membranes (Bio-Rad, Richmond, CA, USA) [61].

The membranes were blocked with Everyblot blocking buffer (Bio-Rad, Hercules, CA, USA) and then incubated with the specific antibody. The following antibodies were used: mouse monoclonal anti-human α-Tubulin; rabbit monoclonal anti-human phospho-p44/42 MAPK (p-Erk1/2; Thr202/Tyr204); rabbit monoclonal anti-human p44/42 MAPK (Erk1/2); goat polyclonal anti-human phospho-IkBα (p-IkBα; Ser32); goat polyclonal anti-human IKbα; rabbit polyclonal anti-human CDK2; goat polyclonal anti-human cyclin A; goat polyclonal anti-human Caspase 3. Primary antibodies were from Cell Signaling Technology, Inc. (Danvers, MA, USA). After washes, the filters were incubated for 1 h at room temperature with horseradish peroxidase-conjugated secondary antibody (Euroclone, Milan, Italy). The membranes were then stained using a chemiluminescence system (Amersham ECL, Cytiva, MA, USA) and then acquired with ChemidocTM MP System (Biorad). Immunoreactive bands were quantified with Imagelab analysis software (version 6.1, Bio-Rad). The membranes immunoblotted with phospho-antibodies (pERK1/2, pIkbα) were stripped and reprobed with antibodies against the total form of the proteins (ERK1/2, IkBα) and then with tubulin as a loading control.

### 4.13. Statistical Analysis

Statistical analyses were performed with Prism 6.0 software (GraphPad, San Diego, CA, USA). Data were expressed as means ± SD and analyzed for significance via a 1- or 2-way ANOVA for independent groups, followed by Tukey’s post hoc correction for multiple comparisons. Values of *p* < 0.05 were considered statistically significant.

## Figures and Tables

**Figure 1 ijms-25-00252-f001:**
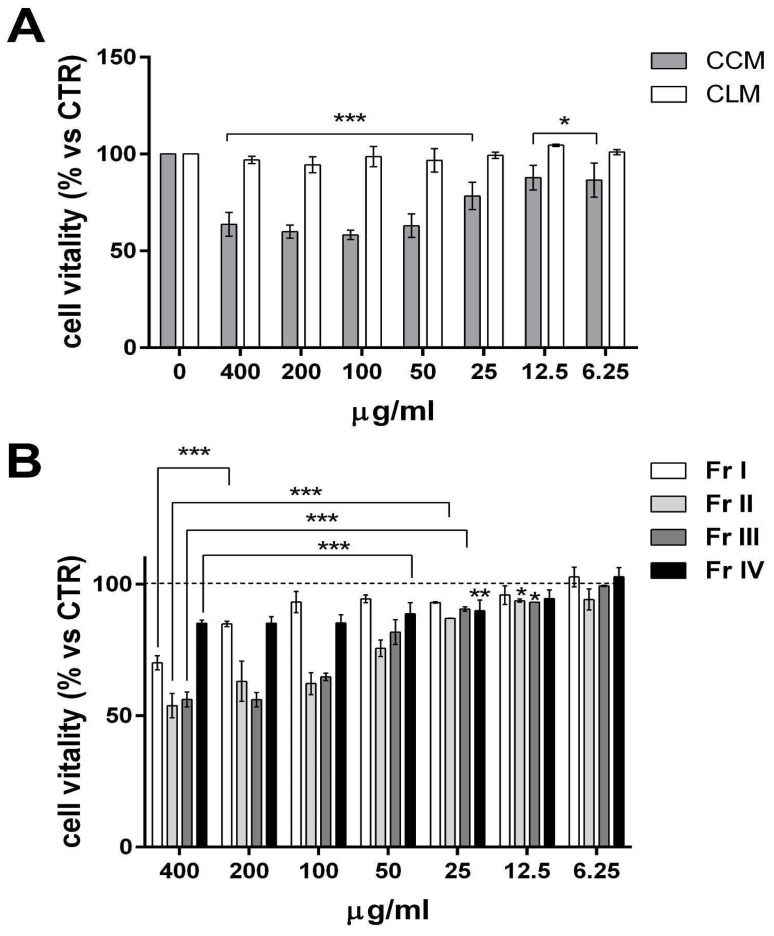
CCM whole extract and its isolated fractions II and III affect AGS cell vitality. (**A**) Histogram showing the viability of AGS cells, expressed as a percentage versus the control condition (dashed line, 100%), treated with a range of concentrations of CCM and CLM extracts (6.25–400 μg/mL) for 24 h by means of an MTT assay. (**B**) Histogram showing the viability of AGS cells treated with the same range of concentrations of CCM fractions I–IV (6.25–400 μg/mL) for 24 h. Results are expressed as means ± SD of 3 independent experiments performed in technical replicates. (ANOVA; * *p* < 0.05, and *** *p* < 0.001).

**Figure 2 ijms-25-00252-f002:**
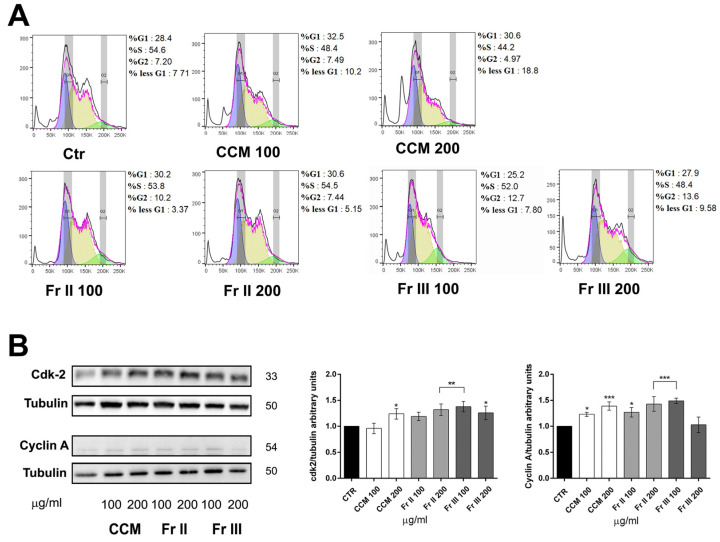
Figure showing that CCM extract and active fractions II and III affect the cell cycle inducing cdk2 and cycA expression. (**A**) Histograms of exponentially growing AGS cells treated with two selected concentrations (100 and 200 μg/mL) of CCM extract and fractions II and III for 24 h. Harvested cells were stained with propidium iodide (PI) and subjected to flow cytometry analysis. The histograms report the cell cycle distribution from univariate modeling via FlowJo software analysis (magenta line). The percentage of the cells in each phase of the cell cycle (G1 = blue peak; S = yellow peak; G2 = green peak) along with the percentage of sub-G1 events is reported on the right of each graph. A representative experiment out of three with overlapping results is shown. (**B**) Immunoblot of AGS cells treated as above with CCM and fractions II and III at the indicated concentrations.. For each blot stripped and reprobed with the different antibodies shown, tubulin was used as a loading control. The blots are representative of 3 independent experiments with similar results. The histograms on the right show the quantitative analysis in arbitrary units expressed as means ± SD of the results from the 3 independent experiments. (ANOVA; * *p* < 0.05, ** *p* < 0.01, and *** *p* < 0.001 vs. control).

**Figure 3 ijms-25-00252-f003:**
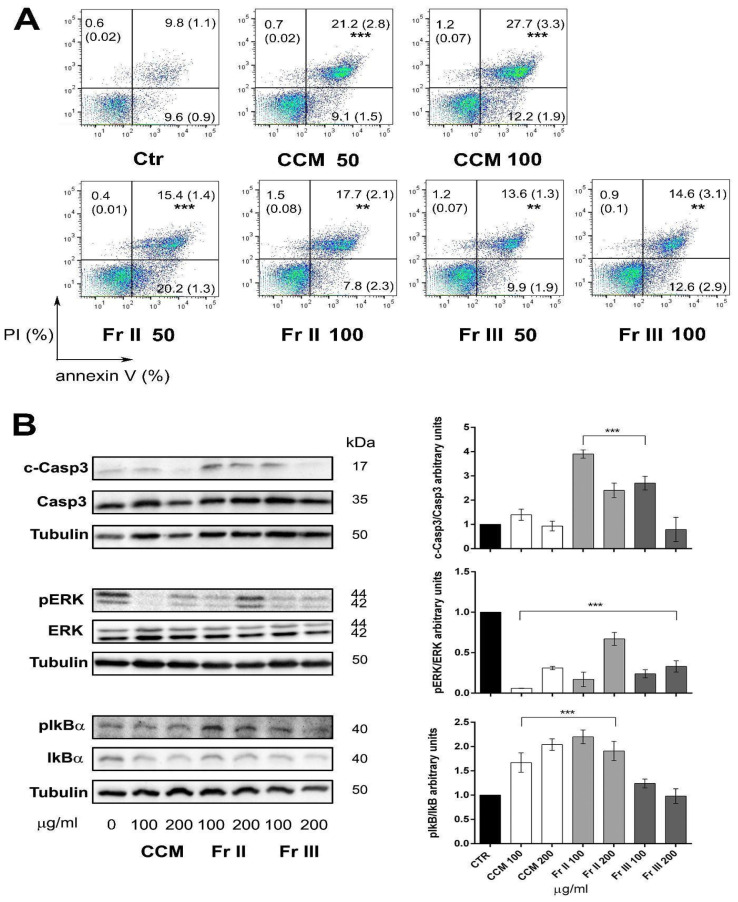
Figure showing that CCM extract and active fractions II and III induce AGS cells apoptosis. (**A**) Pseudocolor density plots showing quantitative analysis of Annexin V-positive (early apoptotic, lower right quadrant), PI-positive (necrotic, upper left quadrant) and Annexin-PI-double-positive (late apoptotic, upper right quadrant) cells (increased cell density from blue to green). The experiment shown is representative of 3 independent experiments that were carried out in duplicate. The mean percentages with SD in parentheses are expressed on each dot blot along with significance. (**B**) Figure showing immunoblot of AGS cells treated for 24 h with CCM and Fr II and III at the indicated doses. Tubulin was used as loading control for each immunoblot that was stripped and reprobed with the different total and phosphorylated antibodies. The blots in the picture are representative of 3 independent experiments with similar results. The histograms on the right show the quantitative analysis in arbitrary units expressed as means ± SD of the results from the 3 independent experiments. (ANOVA; ** *p* < 0.01, and *** *p* < 0.001 vs. control).

**Figure 4 ijms-25-00252-f004:**
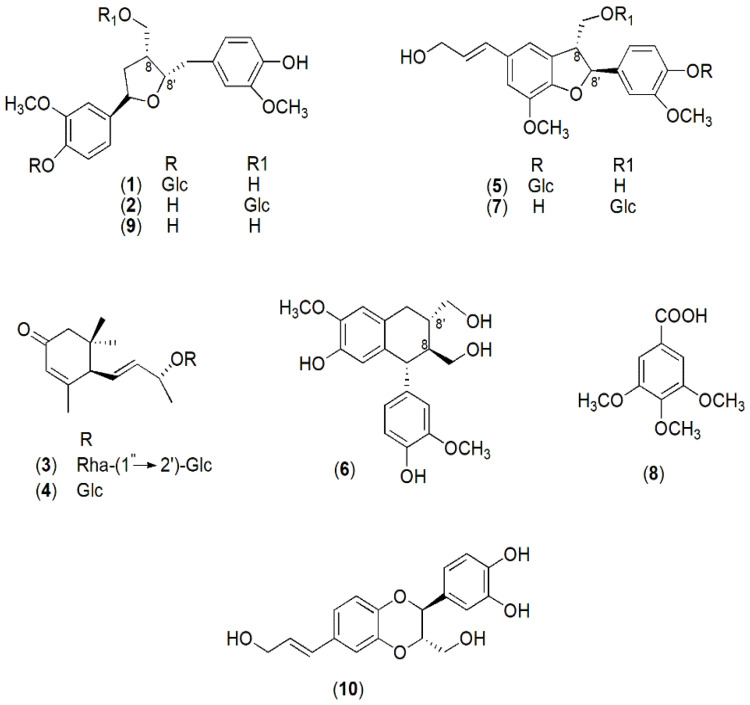
Chemical structures of compounds **1**–**10** isolated from Colocasia corm extract (CCM). Glc: β-d-glucopyranosyl; Rha: α-l-rhamnopyranosyl.

## Data Availability

Data is contained within the article and Appendix A.

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
