# Peer review of "Activity of Colocasia esculenta (Taro) Corms against Gastric Adenocarcinoma Cells: Chemical Study and Molecular Characterization"

_ijms, 2023, doi:10.3390/ijms25010252_

Round 1
Reviewer 1 Report
Comments and Suggestions for Authors
Author Response
Thank you for your input and suggestions. A list of point-by-point authors' responses is attached.

Reviewer 2 Report
Comments and Suggestions for Authors
The manuscript "Activity of Colocasia esculenta (Taro) Corms Against Gastric Adenocarcinoma Cells: Chemical Study and Molecular Characterization" investigates the cytotoxic activity of C. esculenta corms and leaves on AGS cells using the MTT assay. Furthermore, the study described the fractionation of a methanolic extract of Colocasia corms using molecular exclusion chromatography into four fractions, which were then examined for their inhibitory effect on AGS cell viability using the MTT assay. This study also reported on the isolation and identification of several physiologically active compounds from biologically active fractions. The experimental protocols are all thoroughly detailed. The work is engaging, and it finally provides significant insights to its readers; nonetheless, the authors must acknowledge and resolve some flaws. To validate the work's reliability, I propose that the NMR spectra, especially the 2 dimensional spectra be published as extra material.
Author Response

(The authors gave the same response as above.)

Reviewer 3 Report
Comments and Suggestions for Authors
Review of manuscript (Manuscript ID: ijms-2689048) entitled „ Activity of Colocasia esculenta (Taro) Corms Against Gastric Adenocarcinoma Cells: Chemical Study and Molecular Characterization” submitted by Tiziana Esposito et al. to Int. J. Mol. Sci.
Authors have identified a novel potential therapeutic agent, i.e. total extract - the whole Colocasia phytocomplex, derived from an edible root of the plant Colocasia esculenta, commonly known as taro, which has demonstrable intriguing anti-tumor activity against gastric cancer. The total extract and isolated fractions II and III antitumor activity was assessed in an in vitro model of gastric adenocarcinoma (AGS cells). This results shown that their affected AGS cell vitality in a dose-dependent manner through the modulation of key proteins involved in cell proliferation, apoptosis, and cell cycle processes, such as caspase 3, cyclin A, Cdk2, IkBα, and ERK. The proposed subject can raise some interest in scientific community. After a little improvement (minor revision), the article qualifies for publication:
1. Figure 2A is not clear - the images are too small and there is no explanation of what each color in the figure means.
2. The abbreviation used for cyclin-dependent kinase 2 is "Cdk2", not "ckd2".
3. All Figure 4 should be correct.
a) Presented common structure for compounds (1), (2) and (9) has wrong a core and also chirality (is: 2R,3S,5S; should be: 2S, 3R, 4R); see website: “https://pubchem.ncbi.nlm.nih.gov/compound/Lariciresinol#section=Structures" for (+)-lariciresinol structure as an example.
b) In a chemical structure of compound 6, (+)-isolariciresinol, all bonds at the chiral centers should be marked to enable identification of which isomer is represented.
4. Why is the first sentence highlighted in yellow in “4.8. Cells" part?
Author Response

(The authors gave the same response as above.)

Round 2
Reviewer 1 Report
Comments and Suggestions for Authors
Comments
The most serious concern throughout is that it is only known that the extract has the ability to induce cell death, and we still have no idea what the active component is. The reviewer believes that this active component should be used to investigate the molecular mechanism, and that the current data are only at the beginning. I look forward to future developments.
In research on cell death, it is best to perform at least a rescue experiment. Typical inhibitors are available at low cost and the experiment is not very difficult. At the bare minimum, evidence such as that when caspase 3 cleavage is suppressed, cell death is also suppressed is necessary.
The author responded that DMSO was mentioned, but the paper simply stated, "(DMSO), used for solubilizing the extract and molecules during bioassays,". Surprisingly, the revised version states that the concentration is "Values from treated cells were normalized against control DMSO tested in the concentration range from 0.4 to 6.25∙10-3%." With this, it is still impossible to determine what is the control (what percentage of DMSO) in each experiment. Even if cell proliferation is not affected, changes in DMSO in this range generally affect protein phosphorylation. Therefore, I would like to request a unified experimental system throughout.
Author Response
Response to Reviewer #1
Comments
“The most serious concern throughout is that it is only known that the extract has the ability to induce cell death, and we still have no idea what the active component is. The reviewer believes that this active component should be used to investigate the molecular mechanism, and that the current data are only at the beginning. I look forward to future developments.”
Authors’ reply: Overall, our data come to the conclusion that there is no single bioactive molecule but it is the phytocomplex of Colocasia rhizomes that as a whole exerts the anti-cancer action shown. The fact that the phytocomplex has greater biological activity than its individual components is common and has been reported for many plant species widely used in phytotherapy and ethnobotanical practices (see ref 47-49 and many others in the literature). Indeed, we evaluated and reported in the manuscript the cytotoxic activity of the three most abundant molecules of the extract that are (+)-isolariciresinol (6), americanol A (10) and dehydrodiconiferyl alcohol-9-O-β-D-glucopyranoside (7). As you can see at line 206: “The chemical markers displayed an inhibitory efficacy only at the highest concentrations (% of inhibition vs control at the 250 μg/ml dose: 6, 22%; 7, 22%; 10, 35%). Notably, these concentrations were several orders higher than the quantity in μg of each molecule found in 100 μg of the total CCM extract reported above.”
“In research on cell death, it is best to perform at least a rescue experiment. Typical inhibitors are available at low cost and the experiment is not very difficult. At the bare minimum, evidence such as that when caspase 3 cleavage is suppressed, cell death is also suppressed is necessary.”
Authors’ reply: We understand the reviewer’s point of view but it is very difficult to perform such experiments in the few days given to us to review the paper. Furthermore, we remain convinced that this is merely a mechanistic and redundant experiment, as three different assays, widely used in the international scientific literature (to cite just a few: Li X, et al. (2018) Cell Cycle Arrest and Apoptosis in HT-29 Cells Induced by Dichloromethane Fraction From Toddalia asiatica (L.) Lam. Front. Pharmacol. 9:629; Cabrera et al. (2015) G2/M Cell Cycle Arrest and Tumor Selective Apoptosis of Acute Leukemia Cells by a Promising Benzophenone Thiosemicarbazone Compound. PLoS ONE 10(9): e0136878; Li, et al. The Anti-Proliferation, Cycle Arrest and Apoptotic Inducing Activity of Peperomin E on Prostate Cancer PC-3 Cell Line. Molecules 2019, 24, 1472), unequivocally demonstrate the activation of apoptosis by Colocasia rhizome extract.
The author responded that DMSO was mentioned, but the paper simply stated, "(DMSO), used for solubilizing the extract and molecules during bioassays,". Surprisingly, the revised version states that the concentration is "Values from treated cells were normalized against control DMSO tested in the concentration range from 0.4 to 6.25∙10-3%." With this, it is still impossible to determine what is the control (what percentage of DMSO) in each experiment. Even if cell proliferation is not affected, changes in DMSO in this range generally affect protein phosphorylation. Therefore, I would like to request a unified experimental system throughout.
Authors’ reply: As required by the reviewer we already specified in the revised R1 version of the manuscript the percentage of DMSO that was used in each experiment. The entire range of DMSO concentrations was used, as usual, in the MTT assay to exclude that DMSO might have contributed to the observed cytotoxic effect. Excluding this, in subsequent experiments we did not use the full range of DMSO percentages, but only its highest percentage, verifying that there was no difference with the untreated control and using it as a control for each experiment. All this has already been specified in the materials and methods of the revised version of the manuscript. We do not think we need to make any further changes in this regard.

Round 3
Reviewer 1 Report
Comments and Suggestions for Authors
The requested rescue experiment is considered mandatory.
Author Response
We are very sorry that our current experimental design did not fully convince you.